# Establishment of a Rodent Glioblastoma Partial Resection Model for Chemotherapy by Local Drug Carriers—Sharing Experience

**DOI:** 10.3390/biomedicines11061518

**Published:** 2023-05-24

**Authors:** Carolin Kubelt, Dana Hellmold, Eva Peschke, Margarethe Hauck, Olga Will, Fabian Schütt, Ralph Lucius, Rainer Adelung, Regina Scherließ, Jan-Bernd Hövener, Olav Jansen, Michael Synowitz, Janka Held-Feindt

**Affiliations:** 1Department of Neurosurgery, University Medical Center Schleswig-Holstein, UKSH Campus Kiel, 24105 Kiel, Germany; michael.synowitz@uksh.de (M.S.); janka.held-feindt@uksh.de (J.H.-F.); 2Section Biomedical Imaging, Molecular Imaging North Competence Center (MOIN CC), Department of Radiology and Neuroradiology, University Medical Center Schleswig-Holstein, UKSH Campus Kiel, Kiel University, 24118 Kiel, Germany; eva.peschke@uksh.de (E.P.); olga.will@uksh.de (O.W.); jan.hoevener@rad.uni-kiel.de (J.-B.H.); 3Functional Nanomaterials, Department of Materials Science, Faculty of Engineering, Kiel University, 24143 Kiel, Germany; mhau@tf.uni-kiel.de (M.H.); fas@tf.uni-kiel.de (F.S.); ra@tf.uni-kiel.de (R.A.); 4Priority Research Area Kiel Nano, Surface and Interface Sciences (KiNSIS), Kiel University, 24118 Kiel, Germany; rscherliess@pharmazie.uni-kiel.de (R.S.); olav.jansen@uksh.de (O.J.); 5Institute of Anatomy, Kiel University, 24118 Kiel, Germany; rlucius@anat.uni-kiel.de; 6Department of Pharmaceutics and Biopharmaceutics, Kiel University, 24118 Kiel, Germany; 7Department of Radiology and Neuroradiology, University Medical Center Schleswig-Holstein, UKSH Campus Kiel, 24105 Kiel, Germany

**Keywords:** partial resection model, local drug delivery systems, glioblastoma, rodent animal model

## Abstract

Local drug delivery systems (LDDS) represent a promising therapy strategy concerning the most common and malignant primary brain tumor glioblastoma (GBM). Nevertheless, to date, only a few systems have been clinically applied, and their success is very limited. Still, numerous new LDDS approaches are currently being developed. Here, (partial resection) GBM animal models play a key role, as such models are needed to evaluate the therapy prior to any human application. However, such models are complex to establish, and only a few reports detail the process. Here, we report our results of establishing a partial resection glioma model in rats suitable for evaluating LDDS. C6-bearing Wistar rats and U87MG-spheroids- and patient-derived glioma stem-like cells-bearing athymic rats underwent tumor resection followed by the implantation of an exemplary LDDS. Inoculation, tumor growth, residual tumor tissue, and GBM recurrence were reliably imaged using high-resolution Magnetic Resonance Imaging. The release from an exemplary LDDS was verified in vitro and in vivo using Fluorescence Molecular Tomography. The presented GBM partial resection model appears to be well suited to determine the efficiency of LDDS. By sharing our expertise, we intend to provide a powerful tool for the future testing of these very promising systems, paving their way into clinical application.

## 1. Introduction

Amongst all primary brain tumors, glioblastoma (GBM) is the most common and most malignant entity. Despite all research efforts and versatile new approaches, the outlook of the affected patients remains poor—giving a life expectance of 15 months using surgery and radio-/chemotherapy [1]. One reason for this dismal prognosis is the distinct therapy resistance [2]. Treatment failure has been attributed to abundant intra- and intertumoral heterogeneity related to the coexistence of multiple tumor subclones, redundant signaling pathways, and an immunosuppressive microenvironment [3]. Furthermore, the highly invasive nature of GBM makes it difficult to achieve complete surgical resection, and even when resection is successful, there is a high likelihood of tumor recurrence [4]. Finally, inadequate drug delivery across the blood–brain barrier (BBB) and systemic side effects due to desired high local doses of chemotherapeutic agents are major challenges [5].

A promising strategy to treat GBM is local drug release. Local drug delivery systems (LDDS) offer several potential benefits over traditional systemic drug delivery for the treatment of GBM [6]. One advantage is that LDDS can deliver therapeutic agents directly to the tumor site, thereby minimizing systemic toxicity and reducing the risk of side effects associated with traditional chemotherapy. Furthermore, LDDS can enable sustained and controlled drug release, which can enhance therapeutic efficacy by maintaining constant drug concentrations in the therapeutic window at the site of the tumor for an extended period [6]. Additionally, LDDS can bypass the BBB, thereby improving drug availability [7]. Finally, the implantable devices can be tailored to individual patient needs, e.g., with regard to the release kinetics of the drug [8]. Nevertheless, only very few systems have found their way into clinical application so far, and the outcome was limited [9,10]. In particular, local inflammatory reactions and very limited drug release have hindered the success of such systems [9,11]. To address this matter, interdisciplinary research groups worldwide have undertaken a major research effort to develop biocompatible, absorbable LDDS with optimized release properties [12]. 

Another challenge in the development of such novel therapy systems is preclinical testing, most often in rodent animal models. The rat, in particular, is the most commonly selected species for research in mammalian neuroscience since rats are of sufficient size for accurate stereotactic localization of distinct brain areas, and they are resistant to infections. Furthermore, inbred strains are commercially available, allowing more uniformity of the animals at reasonable costs. 

Orthotopic models are often employed to generate optimal testing conditions, which, e.g., allow the important influence of the tumor microenvironment. However, these models are complex and require microsurgical expertise. To date, only a few studies have been published whereby local drug carriers were implanted after resection of the GBM and evaluated regarding their effectiveness [9,10]. 

Thus, it was our goal to detail our experience of establishing a rodent GBM partial resection model for the assessment of the administration of chemotherapy by local drug carriers. For the first time, our study provides a detailed overview of the key considerations and workflow involved in the establishment of an in vivo GBM partial resection model for the testing of promising LDDS, including the selection of the appropriate animal species and GBM cell type used for the induction of tumor growth, the microsurgical technique of partial GBM resection and the implantation of a LDDS, the monitoring of tumor growth, and the in vitro and in vivo validation of drug release from the LDDS. By providing a clear and structured step-by-step explanation addressing common challenges, we aim to pass on important experiences to create a good basis for the future testing of local therapy strategies so that these very promising systems can finally find their way into clinical application.

## 2. Materials and Methods

### 2.1. Animals and Animal Care 

In total, 32 male Wistar rats (weighing 450–500 g at the time of surgery; out of own breeding or ex-breeder) and 8 athymic rats (weighing 250–300 g at the time of surgery; Janvier-Labs, RTA des Chênes Secs, Le Genest Saint Isle, France) were used for the experiments. Animals were acclimatized and maintained in animal facilities on a 12 h light–dark cycle, receiving food and water ad libitum. Concerning the animal husbandry of the athymic rats, they were isolated from the rest of the colony under clean but not sterile conditions, in which cages were equipped with air filters (EBECO, Castrop-Rauxel, Germany) and placed in ventilated cabinets (SCANBUR, Karlslunde, Denmark). Special care was taken to use the minimum number of animals required for statistical accuracy following the 3R-principle. Biometric planning was performed using the statistic program G*Power of the University of Düsseldorf (http://www.gpower.hhu.de/ accessed on 30 September 2019).

Animals were (in case of Wistar rats randomly) assigned to different establishment groups as follows: 

1A Establishment of the model (*n* = 20). Aim: Technical practice, determination of optimal cell amount for inoculation; no survival after partial tumor resection and implantation of the LDDS; one animal of this group was used for the Fluorescence Molecular Tomography (FMT) experiment.1B Examination of the tumor progression after partial resection (*n* = 12). Aim: Observing the clinical outcome after partial resection and implantation of the LDDS as well as monitoring the tumor progression around the implant. 1C Establishment in athymic rats (*n* = 8; including *n*_U87_ = 6; *n*_GSCs (glioma stem-like cells)_ = 2). Aim: Proving the feasibility in athymic rats using human GBM cells. 

Figure 1 gives a schematic overview of the trial. As mentioned before, animals belonging to group 1A were taken out of the trial after partial GBM resection.

### 2.2. Cell Culture

The rat glioma cell line C6 (ECACC 92090409) and the human GBM cell line U87MG (ECACC 89081402) were obtained from the European Collection of Authenticated Cell Cultures (ECACC, Salisbury, UK). C6 cells were cultured under non-stem cell conditions in Dulbecco’s Modified Eagle’s Medium (DMEM; Thermo Fisher Scientific, Waltham, MA, USA) supplemented with 10% fetal bovine serum (FBS; Thermo Fisher Scientific), 1% penicillin-streptomycin (10,000 U/mL; Thermo Fisher Scientific), and 2 mM additional L-glutamine (Thermo Fisher Scientific) as described by Flüh et al. [13]. The purity of the C6 cells was routinely proven by immunostaining with cell type-specific markers and by the absence of contamination with mycoplasma. U87MG spheroids (stem-like cells) were established and intensively characterized, as described by Flüh et al. [13], by cultivating U87MG cells in neurosphere medium (50% DMEM, 50% F12 medium (Thermo Fisher Scientific)) containing the following supplements: 2 mM L-glutamine, 0.6% glucose (Roth, Karlsruhe, Germany), 9.5 ng/mL putrescine dihydrochloride (Sigma-Aldrich, St. Louis, MO, USA), 6.3 ng/mL progesterone (Sigma-Aldrich), 5.2 ng/mL sodium selenite (Sigma-Aldrich), 0.025 ng/mL insulin (Sigma-Aldrich), 2 g/mL heparin (Sigma-Aldrich), and 4 mg/mL bovine serum albumin (Thermo Fisher Scientific). The growth factors EGF (epidermal growth factor; PeproTech, Rocky Hill, NJ, USA) and bFGF (basic fibroblast growth factor; ImmunoTools, Friesoythe, Germany) were added at a concentration of 20 ng/mL. The identity of U87MG cells was proven routinely by Short Tandem Repeat profiling at the Department of Forensic Medicine (Kiel, Germany) using the Powerplex HS Genotyping Kit (Promega, Madison, WC, USA) and the 3500 Genetic Analyser (Thermo Fisher Scientific).

Human patient-derived GSCs were generated via dissociation of tumor material obtained by surgical dissection at the Department of Neurosurgery (Kiel, Germany) with approval of the ethics committee of the University of Kiel, Germany, after receiving written informed consent from donors (file reference: D471/15 and D524/17) and in accordance with the Helsinki Declaration of 1975 as described by Flüh et al. [13]. GSCs were cultured under stem-like cell conditions in F12 media supplemented with B27 supplement (Thermo Fisher Scientific), 2 mM L-glutamine, and 1% penicillin-streptomycin (10,000 U/mL). The growth factors EGF and bFGF were added at a concentration of 10 ng/mL. GSCs were characterized by the formation of neurospheres, the ability to survive and proliferate under stem cell conditions, and the ability to differentiate into more mature cells, which was proven by Flüh et al. [13]. The purity of the GSCs was ascertained by immunostaining with cell-type-specific markers and by the absence of contamination with mycoplasma.

Counting of viable cells was performed using a hemocytometer and trypan blue staining (0.4% trypan blue solution, Thermo Fisher Scientific), allowing for the distinction between viable and dead (=trypan blue-stained) cells.

### 2.3. Surgical Procedures

After sedation of the animals (see Section 3.3.2) the eyes were protected with Bepanthen^®^ ointment (Bayer Vital GmbH, Leverkusen, Germany) and the fur on the head of the animal was carefully removed using an electric hair clipper. The surgery took place on a heating mat at 37 °C under the monitoring of oxygen levels (aim: >95%) and heart frequency (aim: 250–400/min). In case of a deviation of the measured values from the targeted levels, intervention (e.g., new positioning, antagonization of sedation or deeper sedation) took place. The narcotized rats were clamped into a stereotactic frame in a prone position to allow a three-dimensional positioning of the Hamilton needle. Then, sterile washing (Kodan^®^, Schülke & Mayr GmbH, Norderstedt, Germany) and covering of the surgical field were performed. 

Tumor cell inoculation: Regarding the inoculation of tumor cells, a sagittal incision along the midline over a length of approx. one centimeter was performed. To achieve hemostasis, H_2_O_2_ followed by Ringer’s solution was applied to the situs. After flushing the wound with the local anesthetic agent lidocaine^®^ (B. Braun SE, Melsungen, Germany), the periosteum was detached from the bone using a raspatory to expose the bregma. The wound was held apart by two clamps placed in the subcutaneous tissue. Three millimeters laterally (left) and one millimeter anterior to the bregma, a borehole was drilled using a spherical head-shaped drill—as used in dentistry. Then, the dura mater was perforated using a 26G cannula. The cell suspensions (up to 2 × 10^5^ C6 cells, up to 4 × 10^5^ U87 stem-like cells, and 2 × 10^5^ patient-derived GSCs in 5 μL volume) were inoculated with a Hamilton syringe. The needle of this 5 μL Hamilton^®^ microsyringe (Geyer GmbH, Hamburg, Germany) was stereotactically inserted into the borehole and manually slowly advanced to a depth of 5.5 mm (advancement speed: approx. 1 mm/min). After a short waiting time, the syringe was withdrawn by 0.5 mm in order to create a reservoir for the tumor cells and the 5 μL of cell suspension was carefully administered at a rate of approx. 1 μL/min. After a waiting time of 5 min, the slow retraction of the syringe took place (1 mm/min). Finally, the borehole was closed with bone wax, and the wound was closed with sutures. 

Partial tumor resection and implantation of the LDDS: Regarding partial tumor resection, a skin incision of 2 cm was performed. After achieving hemostasis, local anesthesia, the introduction of clamps (see tumor cell inoculation), and a craniectomy of 5 × 5 mm was performed. The dura was opened in a star shape using a micro scissor. For a better overview, the dura margins were then resected. In most of the animals, abnormalities of the cortex due to tumor-caused hemorrhage or tumor growth were visible. The tumor was then carefully partly resected using micro scissors, tweezers, and tumor alligator forceps. Materials from human medicine were used to control bleeding (see Section 3.3.2). After an approximately 3–4 mm resection cavity had formed, an exemplary LDDS, containing a reservoir and fabricated as described before, was loaded with Ringer’s solution and introduced [14]. As an exemplary drug delivery system, a macroscopic polydimethylsiloxane (PDMS) matrix (cylindric, 3 × 3 mm, 48% porosity, 10 μL volume) composed of interconnected microchannels for diffusion-controlled drug release was used [15]. Then, the situs was filled with Ringer’s solution at body temperature. The dural opening was closed with TachoSil^®^ (Takeda Pharma Vertrieb GmbH & Co. KG, Berlin, Germany), and the skin was sutured. Finally, the wound was cleaned and disinfected, and MRI was performed (see Section 2.4). 

After finishing all procedures, the animals were placed on a heating mat, and anesthesia was partially antagonized in group 1B and 1C (see Section 3.3.2). Furthermore, the animals of group 1B and 1C received a special postoperative drug regimen (see Section 3.3.2). All rats were tightly observed until the anesthesia had subsided, and were not returned to their cages until orientation and normal mobility had returned. After completion of the trial or if abortion criteria were fulfilled, euthanasia was performed by application of 4-fold sedation directly followed by the perfusion of the animals by heart puncture, installing cold phosphate-buffered saline (PBS, pH 7.4) and 4% paraformaldehyde (PFA).

### 2.4. MRI Measurements

MRI measurements were performed by using a 7T preclinical MRI (BioSpec 70/30, Paravision 360, Avance NEO, Bruker, Germany) equipped with an 86 mm quadrature volume transmit-receive coil and a 2 × 2 channel receive-only surface coil. T2-weighted Turbo RARE (Rapid Acquisition with Relaxation Enhancement) was performed with the following imaging parameters: echo time = 41 ms, repetition time = 3237 ms, resolution = 0.132 × 0.132 mm^2^, field-of view = 35 × 35 cm^2^, slice thickness = 0.5 mm, slices = 15, RARE factor = 8, and scan time = 01:46 min. Before and after contrast agent application, a T1-weighted RARE was used with the following imaging parameters: echo time = 7.04 ms, repetition time = 750 ms, resolution = 0.132 × 0.132 mm^2^, field-of view = 35 × 35 cm^2^, slice thickness = 0.5 mm and scan time = 01:39 min, slices = 15, RARE factor = 2. As a contrast agent, Gadobutrol (Gadovist^®^, 10 μg/kg body weight, intravenous application, Bayer Vital GmbH) was used. 

### 2.5. Release Measurements of an Exemplary LDDS In Vitro

After fabrication of the laser-structured PDMS microchannel matrices with height and diameter of 3 mm and porosity of 48% for fast release kinetics (as described before), the samples were infiltrated with 10 μL of a 3.2 μM IRDye 800CW near-infrared fluorescent dye (Li-Cor Biosciences, Lincoln, NE, USA) solution (in PBS) under low vacuum in a desiccator, leading to the filling of the microchannel network [15]. After carefully rinsing the samples with PBS to remove excess IRDye 800 solution, the release of the dye was measured in 500 μL of artificial cerebrospinal fluid (aCSF) containing 124 mM NaCl, 5 mM KCl, 26 mM NaHCO_3_, 1.3 mM MgCl_2_, 2 mM CaCl_2_, and 10 mM glucose in aqua bidest, pH 7.3. The release of the IRDye 800 was monitored under sterile conditions at 37 °C/5% CO_2_ under continuous planar rotation at 25 rpm using the KM CO_2_-FL small shaker (Edmund Bühler GmbH, Tübingen, Germany). Released amounts of the IRDye 800 were analyzed using a Fluorescence Plate Reader (Tecan Infinite 200Pro, Tecan Trading AG, Männedorf, Switzerland) at an excitation of 770 nm and 800 nm emission after 1, 2, 3, 4, and 72 h, and determined using an external calibration of defined standard IRDye dilutions.

### 2.6. Release Measurements of an Exemplary LDDS In Vivo

In order to determine the release of an exemplary LDDS in vivo, the LDDS was loaded with 10 μL of a 3.2 μM IRDye 800CW near-infrared fluorescent dye solution (in PBS), rinsed with PBS to remove excess IRDye 800 solution, and introduced into the approximately 4 mm-diameter-measuring resection cavity of a partial tumor-resected Wistar rat. The surgeries (tumor cell inoculation, partial resection, and implantation of the LDDS) on the animal were performed according to the procedures described in Section 2.3. Hourly FMT measurements of the animal were carried out over a test period of 4 h using FMT 2500 (PerkinElmer, Waltham, MA, USA). Released amounts of the IRDye 800 were additionally analyzed in 10 μL CSF, which was aspirated from the resection cavity containing the LDDS after 4 h post-implantation directly before the end of the experiment at an excitation of 770 nm and 800 nm emission using a Fluorescence Plate Reader. The concentration of the released dye was determined using an external calibration of defined standard IRDye dilutions.

### 2.7. Statistical Analysis

The data were statistically analyzed using GraphPad Prism 5 Software (GraphPad Software, San Diego, CA, USA). In general, the data are presented as mean ± standard deviation. Statistical analysis was performed by a one-sample *t*-test. Statistical significance is marked with asterisks depending on the *p*-value: * *p* < 0.05, ** *p* < 0.01 and *** *p* < 0.001.

## 3. Results

### 3.1. Selection of a Suitable Animal Model and Cell Lines

In every animal project, one of the first questions that arises is about choosing the right animal model. In order to implant human tumor cells, special immunodeficient animals are needed. However, these are quite expensive and not necessarily needed to establish and train most parts of the desired process. Thus, to keep the costs of establishing the model reasonable, we chose Wistar rats for the first phase. Even though no human cells can be introduced into this type of rat, cost-efficient training of, in particular, the technique of tumor cell inoculation, the microsurgical technique of partial tumor resection, the introduction of the drug delivery systems, and imaging can be performed. 

To induce tumor growth in Wistar rats, the rapidly proliferating rat C6 glioma cell line was used. The resulting tumors were easy to differentiate from the healthy brain parenchyma. Hence, they facilitated the conductance of the partial tumor resection. 

After the surgical techniques had been well established, athymic nude rats were used, allowing for the introduction of human cells. In our study, we decided on human U87MG spheroids and patient-derived glioma stem-like cells.

### 3.2. Tumor Cells

#### 3.2.1. Creation of Standardized Conditions in the Context of Cell Culture

Standardized cell culture conditions provide the basis for reliable results in in vitro and in vivo studies. In order to achieve homogenous tumor growth in the animals, we standardized the cultivation of C6, U87MG cells, and patient-derived GSCs with regard to passaging, subculture, and seeding density prior to the inoculation of the cells (fixed standardized conditions, see Section 2.2). Three days before implantation, the C6 cells were harvested and seeded at a defined number of cells, yielding 80–90% confluency on the day of implantation. The cultivation of the U87MG spheroids followed a 14-day schedule of splitting into single cells and 13 days of subsequent cultivation under stem-like cell conditions. Thereby it was ensured that the U87MG spheroids were split into single cells 14 days prior to implantation. With regard to the patient-derived GSCs, the cultivation followed a 7-day schedule of splitting into single cells and 6 days of subsequent cultivation under stem-like cell conditions; thus, the patient-derived cells were split into single cells 7 days prior to the implantation. Both cell lines and the patient-derived GSCs were used for implantation at a range of 10 subcultures.

#### 3.2.2. Standardized Inoculation Procedure

Cell cultivation and animal surgery often take place at different locations. This necessitates the transport of the cells to the place of inoculation. Transport of the cells (transported as a cell pellet to reduce shear stress of the cells to a minimum) taking place shortly before the tumor cell application appeared to be associated with late-onset (two to three weeks after inoculation) hemorrhages in the brain area of implantation. Three out of four Wistar rats developed significant tumor hemorrhages and only one revealed a marginal tumor hemorrhage (which is expectable during tumor progression). In contrast, if the cells were transported two to three days before the surgery and allowed to grow after transportation, the bleeding was significantly reduced (Figure 2). Only 1 out of 26 Wistar rats developed a moderate tumor hemorrhage—less pronounced than observed before. As expected, small tumor hemorrhages occurred in 11 out of 26 Wistar rats, and 14 animals showed no signs of tumor hemorrhage in the MRI. Hence, this strategy was used for all further experiments in our study, including all athymic animals.

Another question that arises in the context of tumor cell inoculation into brain tissue is which dispersant should be used for the cells. A large number of physiological solutions are available, including standard cell culture-compatible solutions such as PBS and Ringer’s solution. To check the influence of the dispersant on the viability of the cells to be inoculated, we carried out an in vitro cell counting of viable cells for C6, U87MG cells, and patient-derived GSCs. We dispersed 1 × 10^5^ cells in a volume of 10 μL PBS or Ringer’s solution, respectively, and the total volume was drawn up into the Hamilton syringe (26sG needle), slowly ejected at the speed used for the in vivo application, and counted. There was no significant difference in the number of ejected cells for PBS or Ringer’s solution (Table 1).

However, the cell counting test showed a partly clear discrepancy between the number of cells taken up in the syringe (100,000) and the number of cells subsequently released. This difference was particularly evident for U87MG cells (which are larger than C6 cells) and least evident for the patient-derived GSCs. The reason for the difference may lie in the size and, as a result, increased shear forces or the attachment of the cells to the glass surface of the Hamilton syringe. By using a Hamilton syringe and needle with a larger internal diameter (26G), the difference between the drawn and released cells was significantly reduced (Table 2), indicating that the internal diameter is a critical parameter that has to be adapted with regard to the size of the respective cell type used for the inoculation. In order to take the discrepancy of drawn and released cell counts as a source of error into account, standardized cell counts were carried out before the inoculation of the tumor cells for C6, U87MG, and patient-derived GSCs, and the average difference between the number of drawn and of released cells was calculated and added to the total number of cells used for the preparation of the cell suspension used for inoculation.

To avoid flushing of the cell suspension and a suction effect, which would cause a spread of tumor cells in the puncture channel, the administration of the cells and retraction of the Hamilton needle were performed very slowly (administration speed: 1 mL/min, retraction speed: 1 mm/min; see Section 2.3). Despite this procedure, we regularly observed tumor growth in the vicinity of the puncture channel. In 2 out of 32 Wistar rats and 3 out of 8 athymic rats, no sufficient tumor growth appeared. In athymic rats, no sufficient tumor growth was only observed using U87MG cells.

### 3.3. Laboratory Animals

#### 3.3.1. Creation of Standardized Conditions for Laboratory Animals

To standardize the animal model, we chose rats of the same age (6 weeks +/− 6 days at time of inoculation), weight (Wistar rats: 450–500 g; athymic rats: 250–300 g;), and sex (male). Despite our efforts to achieve reproducible cells, injections, and animals, the tumor growth was heterogeneous (anterior–posterior diameter after 21 days of inoculation was 4.4 ± 1.7 mm in Wistar rats and 1.3 ± 1.1 mm in athymic rats, measured with MRI). In this study, we chose to perform partial resection if the tumor diameter was >3 mm, which was reached, on average, after 21 days for Wistar rats and 35 days for athymic rats. A tumor diameter larger than approximately 6.5 mm tended to result in less tolerance towards anesthesia. Exemplarily MRI monitoring of the anterior–posterior tumor diameter among Wistar rats belonging to group 1B (animals that underwent partial resection and LDDS implantation and were observed concerning their survival and tumor progression around the implant) was undertaken, and the results are shown in Figure 3. 

To ensure comparable tumor sizes, we reverted to monitoring the tumor growth via MRI. The use of MRI was essential to assess the tumor size before and after partial resection, regrowth, and the placement and therapeutic effect of the LDDS. Tumor growth and the size after partial resection are parameters that must be considered, in addition to the sole survival of the animals, to determine the efficiency of a therapy. 

#### 3.3.2. GBM Partial Resection and Perioperative Management

The partial tumor resection and introduction of LDDS were a challenge even with existing surgical expertise. On the one hand, the anatomical structures of the animals are small. On the other hand, not all of the instruments used in human medicine (e.g., neuronavigation, surgical instruments in rat-size) are available in this setting. Eventually, we settled for the following protocol:

First, the animals were sedated using intraperitoneally applied Ketamine and Medetomidine. For the Wistar rats, 80 mg/kg body weight Ketamine and 0.5 mg/kg body weight Medetomidine were used. For the smaller and lighter athymic rats, the dose was reduced by 20%. If the surgical tolerance was not reached using this dosage (meaning animals still felt pain), up to 20% of the diluted drug volume (1:5 dilution with NaCl) was additionally carefully applied by injection under constant monitoring. To prevent hypothermia, the animals were placed on a heating mat throughout the duration of anesthesia (Trixie 76087 Heizmatte, 28 × 45 cm, 32 W, TRIXIE Heimtierbedarf GmbH & Co. KG, Tarp, Germany). In addition, the blood oxygenation level and heart frequency were monitored (MauseST AT Jr, TSE Technical & Scientific, Bad Homburg, Germany). Concordant with human brain tumor resections, the partial tumor resection was carried out using an operating microscope (Stemi 305 KMAT, Carl Zeiss Microscopy GmBH, Oberkochen, Germany) together with an additional cold light source with flexible, two-arm gooseneck light guides for optimal illumination of the operating field (CL 6000 LED, Carl Zeiss Microscopy GmBH). After skin incision, a left frontal craniectomy, including the initially drilled borehole used for tumor cell inoculation, was performed using an electric drill (DGT001462, DEDAKJ, Guangdong, China). Care was taken not to open the venous sinus by keeping a distance to the sagittal suture representing an anatomic marker of the midline and the subjacent venous sinus. Afterwards, the dura was opened and resected (see Section 2.3), and the tumor tissue was exposed using microsurgical instruments such as micro forceps, micro scissors, and a micro dissector, as well as extra small brain cotton pads. By using moistened extra small brain cotton pads, the layer of preparation created by the microsurgical instruments was stabilized. Moistening was achieved by soaking the cotton pads in sterile Ringer’s solution straight before usage. After mobilization of the grayish and glassy-appearing tumor tissue using the micro dissector, the tissue was resected with tumor alligator forceps (Yasargil, Ø 3 mm, 22 cm, Medicon eG, Tuttlingen, Germany). The alligator forceps allowed for the safe and fast excision of tumor tissue without exposing the tumor tissue to strong mechanical forces. Due to the soft texture of the tumor tissue, resection using solely micro forceps and micro scissors turned out to be time-consuming and led to more traction with respect to the surrounding tissue, causing hemorrhages and swelling. We refrained from using an aspirator (as it is standard practice for human surgery) during the tumor preparation and resection to avoid losing valuable tissue due to suction effects. The aim of our partial tumor resection was to reduce the tumor size until a resection cavity of approximately 3–4 mm had formed. This size of the resection cavity allowed uncomplicated implantation of our exemplary LDDS without raising intracerebral pressure. Likewise, this partial resection permitted a rapid examination of the effects of the locally applied anti-tumor agent on the surrounding tumor tissue. A macroscopic, apparent complete resection is technically feasible by the methods described above but requires a longer period of observation until a sufficient tumor relapse has formed around the implant. Besides higher costs concerning animal housing and imaging, the release dynamics of the used LDDS must match this time frame. 

In our study, the outcome of the surgery was not affected by residual anterior–posterior tumor diameter <1 mm, as found upon contrast-enhanced MRI (*n* = 8 of 12 Wistar rats), indicating a good tolerance of partial and complete resection in the animals. 

Before placing the LDDS in the resection cavity, meticulous hemostasis has to be achieved. Dealing with bleeding during partial tumor resection was a particular challenge. The use of bipolar forceps in the animal’s brain can not only cause neurological damage but also falsify the test results. The tumor resection margins represent a critical tissue layer, especially when assessing a local therapeutic strategy. The absorption of the locally applied drug can be limited by obliterating large parts of this area. In contrast, inadequate hemostasis can, in turn, lead to a life-threatening hemorrhage, causing midline shift/raised intracranial pressure, hydrocephalus, and, potentially, a reduction in the release of drugs by the LDDS. To address both points, we used hemostatic material from human medicine. The use of the absorbable hemostats Tabotamp^®^ (network of oxidized, regenerated cellulose; Johnson & Johnson Medical GmbH Ethicon Deutschland, Norderstedt, Germany) and GELITA-SPON^®^ (gelatin sponge; Johnson & Johnson Medical GmbH Ethicon Deutschland), as well as rinsing with Ringer’s solution at body temperature, allowed us to successfully control intracerebral bleeding during the partial resection. Tabotamp^®^ and GELITA-SPON^®^ were removed after hemostasis was achieved. In the rare case of intraoperative brain swelling, the brain parenchyma was rinsed with slightly cooled Ringer’s solution. To prevent dislocation of the implanted LDDS and, in particular, reduce the risk of CSF leakage and consecutive wound healing disorders, the dura was closed using TachoSil^®^. Closure of the dura is important for local therapies as wound healing would also be massively disrupted if there were to be direct contact with the therapeutic agent. In addition, adequate closure of the subcutaneous tissue and cutis is also required. For this reason, we used a simple interrupted suture, which prevailed against intracutaneous sutures and fibrin glue due to higher mechanical resilience and better tolerance by the animals. Figure 4 shows exemplary pictures of the partial tumor resection before implantation of the LDDS.

After surgery and postoperative imaging, the anesthesia with Medetomidine was promptly antagonized using Atipamezole (1 mg/kg body weight, 100 μL subcutaneous administration). Extending the duration of anesthesia to avoid direct postoperative stress in the wake-up period with an increased risk of postoperative bleeding or seizures was refrained from in favor of better clinical monitoring options and a risk-reduction of complications caused by anesthesia, e.g., aspiration. Close clinical monitoring by a veterinarian enabled us to identify and handle complications early (e.g., seizures or pain). The complication of a seizure that occurred immediately after waking up from anesthesia, which was observed only once in our study, was counteracted by the administration of a benzodiazepine (Midazolam, 0.5 mg/kg body weight, intravenous application). We did not see an indication for the prophylactic administration of anticonvulsants, due to the small number of animals involved. To prevent cerebral edema after surgery, the animals received glucocorticoids according to human medicine. Dexamethasone (0.5 mg/kg body weight, intravenous administration) was administered right away and again at 4, 24, and 48 h after surgery. To control the pain, we administered short-term and long-term drugs as follows: Metamizole (100 mg/kg body weight, subcutaneous administration) was administered right away after surgery, whereas Carprofen (10 mg/kg body weight, subcutaneous administration) was given 4, 24, and 48 h after. Furthermore, the drinking water of the animals was supplemented with Tramadol (0.5 mg/mL) for five consecutive days after surgery. Reduced food uptake after surgery was addressed by subcutaneously injecting glucose (3 mL, 5%, *v*/*v*, aqueous solution) right after surgery and 4 h later. 

If the weight loss exceeded 20%, clear erythema or edema of the wound, recognizable pain (assessment by “grimace scale”), or considerable neurological disorders had occurred, the affected animals were removed from the experiment. These demolition criteria show that a single consideration of the survival of the animals is not sufficient to assess the effectiveness of local therapy. Furthermore, for a valid evaluation of the obtained results, it is of great importance to include a sufficient amount of control arms in the trial, e.g., animals that only undergo partial resection without LDDS implantation, receive an unloaded implant after partial resection, or are solely treated with systemic therapy depending on the particular question. 

### 3.4. Testing the Release of Local Drug Delivery Systems

#### 3.4.1. Release of an Exemplary Local Drug Delivery System In Vitro

Before the effectivity of a LDDS loaded with a therapeutic agent can be tested in animal experiments, the release properties of the system should be precisely known to ensure a reproducible biological effect. Here, in vitro tests are well suited, allowing one to determine the release kinetics of the implant under well-controlled conditions with fewer interfering variables. To test our model, we chose a macroscopic PDMS matrix (cylindric, 3 × 3 mm) composed of interconnected microchannels for diffusion-controlled drug release without a reservoir [15]. The microchannel matrix with 48% porosity allows for the loading of 10 μL of the dissolved drug via infiltration. To test the release properties of this exemplary drug delivery system in vitro, the microchannel matrix was loaded with a fluorescent dye (see Section 2.5 for details), allowing for the in vivo detection of release. The release profile of the dye is shown in Figure 5. Strikingly, the dye was continuously released from the implant and exhibited the desired relatively fast initial release. 

#### 3.4.2. Release of an Exemplary Local Drug Delivery System In Vivo

Despite the successful results from the in vitro experiment, an in vivo test of the release is needed since blood contamination of the CSF in the resection cavity and close contact of tissue to the implant can significantly limit the release by occluding the microchannels of the LDDS. To test the release of the exemplary drug delivery system in vivo, the system was loaded with 10 μL of a 3.2 μM solution of the IRD 800 dye according to the in vitro test. The loaded implant was then introduced into the resection cavity (ca. 4 mm diameter) of a partial tumor-resected Wistar rat. The surgeries (tumor cell inoculation, partial resection, and implantation of the LDDS) of the animal were performed according to the procedures described in Section 2.3. Hourly FMT measurements of the animal were carried out over a test period of 4 h. FMT showed a steady increase in fluorescence in the resection cavity—likely a sign of a release in vivo (Figure 6). These findings were further supported by the detection of the dye in the CSF of the resection cavity using a fluorescence plate reader at an excitation of 770 nm and emission of 800 nm. For this purpose, 10 μL microliters of CSF were aspirated from the resection cavity containing the LDDS after 4 h post-implantation, directly before the end of the experiment. About 0.105 μM dye was found in the resection cavity, which is in line with the released dye concentration observed in the in vitro release study after 4 h (c_in vitro_ = 0.116 μM, see Figure 5). These findings suggest that the model presented here (the inoculation of the tumor, partial resection, and LDDS placement) allows for the investigation of LDDS release in vivo.

### 3.5. Monitoring of the Animal Experiment Using MRI

In order to monitor tumor growth, document the postoperative status, and assess the amount of residual tumor after resection, regular (mostly weekly) MRIs were carried out (Figure 7). During the MRI, the sedated animals were placed on a heating mat and monitored by breathing rate. The standardized positioning of the animals (prone position, head first in a straight position), standardized selection of adequate MRI sequences (details given in Section 2.4), and standardized positioning of the scanned slices were decisive. In order to better identify the tumor tissue, an intravenous contrast agent was administered, and a (residual) tumor was thereby defined as an intensely enhancing mass on T1-weighted images.

### 3.6. Further Tissue Processing

Depending on the planned examination technique (e.g., gene expression analysis or staining), the tissue must be treated differently immediately after removal from the animal. Hence, it is essential to plan which specific examination is to be carried out before the tissue is removed.

To achieve the optimal conditions for gene expression analysis, the tissue can be stored in liquid nitrogen immediately after removal. For staining cryosections (e.g., 3, 3′-diaminobenzidine or immunofluorescence double staining), placing the brain tissue in 4% PFA immediately after removal for 24 h, incubation in 15% sucrose for 6–12 h and finally, 30% sucrose for 24 h is advantageous concerning tissue preservation. After this, the tissue or the respective sections can be stored at −30 °C. Alternatively, the tissue can be stored in paraffin. Here, the freshly extracted tissue should be first stored in 4% PFA for 24 h, followed by storage in 1% PFA until embedding in paraffin. Storage in paraffin permits the staining of tissue sections as well as gene expression analysis by using commercially available kits as described before [16]. 

In the framework of tissue analysis, different regions of the brain are of great interest (e.g., tumor, peritumoral area, and areas further away from the tumor tissue). To simplify a selective examination of these distinct areas, a dissection of the brain into sections before storage might be useful. Brain matrices (e.g., stainless-steel brain matrix with coronal slices, 0.5 mm, for rats 300 to 600 g; RBMS-305C, Kent Scientific, Torrington, CT, USA, Figure 8) offer the advantage of a standardized dissection of the tissue, allowing for separate storage and, as a result, the examination of each section and faster detection of the region of interest, respectively.

The optimal conditions for dissecting the brain into sections can be reached after preserving the brain for 24 h in 1% PFA in the case of paraffin sections or after 24 h in 30% sucrose in the case of cryo-conservation. Serial sections can then be made from the respective brain slices.

## 4. Discussion

Despite all research efforts, there are still no curative treatment strategies for the most common and, at the same time, most aggressive primary brain tumor—GBM [1]. The use of LDDS provides a promising strategy to improve this dismal prognosis; nevertheless, the results of the few clinically used systems are not convincing [17,18]. In order to be finally able to apply new advanced local therapy systems clinically, adequate testing in animal studies is inevitable. The implementation of animal models, some of which are very complex, is decisive for the quality of the data collected. An exchange of experiences is therefore not only crucial in order to reduce the number of included animals and their suffering but also to generate high-quality data that will form the basis for clinical applications of LDDS.

### 4.1. Selection of Suitable Animal Models and Cell Lines 

The most common hosts of C6 cell line for an in vivo study are immunocompetent Wistar rats [19]. C6 cells grown intracerebrally in Wistar rats were found to form tumors with several characteristics of malignant glioma, including nuclear pleomorphism, high mitotic index, foci of tumor necrosis and intratumoral hemorrhage, and parenchymal invasion showing characteristics closer to GBM than tumors grown in other rat strains [20,21]. Although widely used to establish a rat glioma model, tumors derived from C6 cells do not reflect the same heterogeneity, immunocytochemical features, and invasiveness as observed in human malignant gliomas [22]. In addition, the major disadvantage of adherent serum-growing cultures are genomic alterations not corresponding well to the genotype of the original tumor [23]. Therefore, athymic nude rats were used after the surgical technique of partial GBM resection was well established, and tumor growth in these rats was induced by implantation of human U87MG spheroids or patient-derived GSCs. Glioma spheres have been shown to more closely reiterate the molecular makeup of the original patient tumor [24,25]. In addition, spheres were shown to be highly tumorigenic and, more importantly, show variable amounts of angiogenesis and extensive invasion in vivo [24]. U87MG has been shown to exhibit profuse neovascularization and has been widely used to study GBM angiogenesis [26]. In contrast, patient-derived orthotopic xenograft models allow personalized drug efficiency tests in single patients and are genetically stable. In addition, the original tumor architecture and histological characteristics are preserved [27]. Likely related to age-dependent changes in immunocompetence, previous studies have documented increased incidences of tumor regression in nude rats [28,29]. These athymic rats reveal an innate lack of normal thymus and, as a result, functionally mature T cells. Since effector T-cell infiltration was shown to positively impact the survival of GBM patients, this factor should be considered [30]. Even if animal glioma cell lines introduced into Wistar rats might offer the advantage of glioma-like immune system evasion strategies, the transferability of this animal model to human conditions is limited [31]. Therefore, the choice of animal species is absolutely dependent on the specific question of the respective project, making athymic rats the most well-suited model for the examination of the efficiency of the LDDS in our study. 

### 4.2. Tumor Growth Behavior

An important requirement to obtain high-quality data is the standardization of the test conditions. Despite profound standardization of the test conditions, we observed clear differences with regard to tumor growth comparing the Wistar rats and athymic rats, respectively. Given that animals are biological systems, this is not fully surprising and ultimately a phenomenon also described in other studies [18,19]. Our observation that some of the animals did not develop a tumor after the inoculation of tumor cells can be found in the literature as well [32].

In addition to the different tumor growth rates, different tumor configurations were observed, e.g., revealing tumor tissue, in particular, at the puncture channel of some animals. This was also observed in previous studies [33]. The speed of inoculation and subsequent removal of the needle, which, in some cases, varies significantly between published studies, certainly influences the distribution of the tumor cells. Nevertheless, spreading tumor cells in the puncture channel was not avoided even with a comparatively slow inoculation technique (e.g., Ozeki et al. applied the cells three times as fast and removed the Hamilton needle twice as fast in comparison to our study [34]). 

### 4.3. GBM Partial Resection and Perioperative Management

With regard to the surgical resection techniques used, studies published to date also differ significantly. Several studies removed the tumor tissue by aspiration, e.g., Ozeki et al., who examined a combination therapy of surgical tumor resection with the implantation of a drug containing hydrogel in a C6 rat glioma model, used needle aspiration (Ø = 1.5 mm) after tumor cell inoculation [34,35,36]. Since the cited studies focused on the efficiency of treatments, a straight comparison of the resection techniques is difficult. Doubtless, a resection by aspiration permits a fast and minimally invasive resection approach; however, subsequent histological examination of the tumor tissue is impeded. Another possibility of resection found in the literature is a tissue punch [37,38]. Bastiancich et al., for instance, used a three-millimeter-diameter biopsy punch inserted four millimeter deep and twisted for 15 s to cut the brain/tumor tissue after the inoculation of C6 or 9L cells in rats [38]. Once withdrawn, the tumor and brain tissues were aspired using a vacuum pump. Altogether, no body weight loss, change in behavior, or neurological symptoms were observed after surgical resection of the tumors, indicating that resection by biopsy punch is a safe surgery approach [38]. However, an individual resection of the tissue that considers the spatial extent of the tumor, as performed in our study by using conventional microsurgical resection, is not possible with a biopsy punch. In addition, using a punch only allows the tumor to be located superficially since, otherwise, too much healthy tissue would be removed. Many studies, as in most of the studies mentioned above, use a superficial tumor growth by injecting tumor cells in a depth of 2–3 mm since this procedure allows for simpler (partly) resection with a consequently reduced complication rate [35,36]. Nevertheless, this location does not reflect the human conditions and, as a result, limits the informative value of the study. Further, a superficial location of the drug delivery system might lead to the fast removal of the drug by the CSF flow. One difficulty, especially when resecting deep-seated tumors, is hemostasis with limited visibility of the operating area. In our study, as in previous studies, this was one major reason for animals to be removed from the experiment at an early stage—shortly after surgery [39]. 

Another challenge when investigating intracerebral drug delivery systems is the closure of covering tissue layers. A single suture of subcutaneous and skin tissue, as performed in previous rodent glioma resection models, might impair wound healing in the setting of local drug delivery [40]. Due to the size of the craniectomy, a simple closure of the skull bone, e.g., with bone wax, is difficult. The insertion of a gelatin sponge does not yield watertight closure, as is required in this particular setting [41]. Other methods, such as the reinsertion of the resected skull bone and fixation with cyanoacrylate glue, bear the risk of dislocation of the glue in the resection cavity and restriction of release and uptake of the locally administered drug [42]. The duraplasty with TachoSil^®^, as used in our study, is cost-intensive but has the advantage of yielding a watertight seal. In addition, the material is stretchable in contrast to the reinserted bone and, as a result, can counteract an increase in intracranial pressure. One major cause of increased intracranial pressure is brain swelling. The risk of this dangerous intra- and postoperative complication is especially raised due to the insertion of a LDDS. The size of the partial resection cavity has to be adapted to the size of the LDDS used to allow for the uncomplicated implantation of material without raising intracerebral pressure. In order to avoid the swelling-related clinical deterioration of the animals caused by the surgical procedure itself and by the introduction of the LDDS into the resection cavity, we additionally used a drug edema prophylaxis with dexamethasone. Due to the successful edema control by this drug, an additional administration of mannitol, as used by Denbo et al., was not necessary for our study [35]. Furthermore, since we observed that a tumor diameter greater than approximately 6.5 mm tended to result in worse tolerability of anesthesia, it is evident that, presumably, the introduction of a LDDS with a maximum size of 4–5 mm is tolerable by the animals with the formation of a resection cavity of 5 mm. The tolerability of such a LDDS is not only linked to its sheer size but also to the material itself. It is advantageous if softer materials such as hydrogels are used, as they allow for the conformational adhesion of the surrounding tissue, which facilitates better drug penetration into the brain tissue. Additionally, biodegradable systems are favorable since the mechanical stress of foreign material to the surrounding tissue decreases over time, and further surgical procedures to remove the LDDS can be avoided [6]. 

### 4.4. Using of Imaging Techniques

To monitor tumor growth and quantify the tumor tissue remaining after resection, different techniques can be used (for example, bioluminescence [35,43]). In our study, we deliberately used MRI, as this technique is particularly applied in human glioma imaging to determine residual tumor tissue after resection [44]. MRI provided superb soft tissue contrast and allowed us to monitor all essential stages of the project (injection, growth, resection, residual tumor, LDDS placement). Although it was not observed here, some LDDS materials may cause MRI artifacts and should be checked first. While MRI provides easy access to the apparent (contrast-enhancing) tumor size, it is well known that size alone is not a good measure for therapy response. For a more detailed picture and early response detection, metabolic MRI, e.g., using hyperpolarized MRI, appears promising [44,45]. FMT was well suited to monitor the release properties of the LDDS used. Although it should be noted that this approach is limited to small animals, has limited spatial resolution, and that the release of the drug may not be the same as the release of the dye. Still, the information gained was very valuable, providing a non-invasive measurement for the release properties. 

The model established in our study raised the claim to resemble human conditions to allow for greater transferability into a clinical setting. Of course, this model is complex and cost-intensive. Thus, it is even more important to share experiences and benefit from them. 

Altogether, our established rodent GBM partial resection model offers a valid basis to test local chemotherapy released by intracavitary located drug carriers.

## 5. Conclusions

LDDS represent a promising treatment strategy concerning the most common and malignant brain tumor—GBM. However, the further development of this therapy approach requires complex preclinical testing, which contributes to the fact that only a few of these systems have found their way into clinical practice. Hence, passing on experience gained in this setting is of great importance. By providing a clear and structured step-by-step explanation addressing common challenges when trying to establish an in vivo GBM partial resection model for the testing of promising LDDS, including the selection of the appropriate animal species and GBM cell type used for the induction of tumor growth, the microsurgical technique of partial GBM resection and the implantation of a LDDS, the monitoring of tumor growth, and the in vitro and in vivo validation of drug release from the LDDS, we aim to pass on important experiences to create a good basis for the future testing of local therapy strategies so that these very promising systems can finally find their way into clinical application. 

## Figures and Tables

**Figure 1 biomedicines-11-01518-f001:**
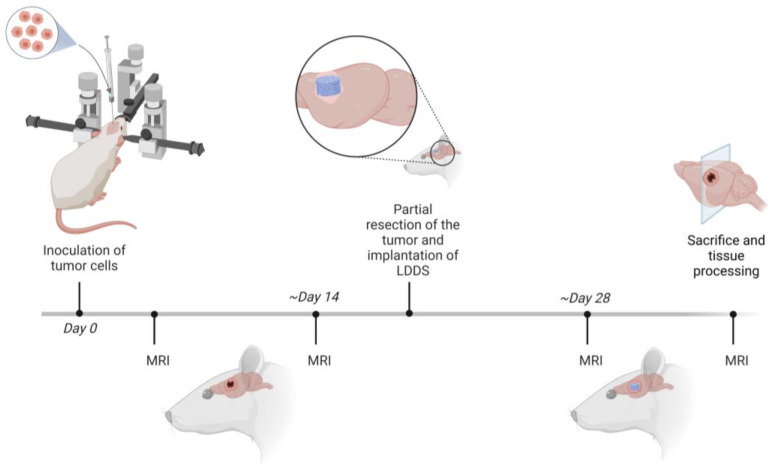
Schematic overview of the animal experiment (created with biorender.com accessed on 7 March 2023). MRI: Magnetic Resonance Imaging; LDDS: local drug delivery system.

**Figure 2 biomedicines-11-01518-f002:**
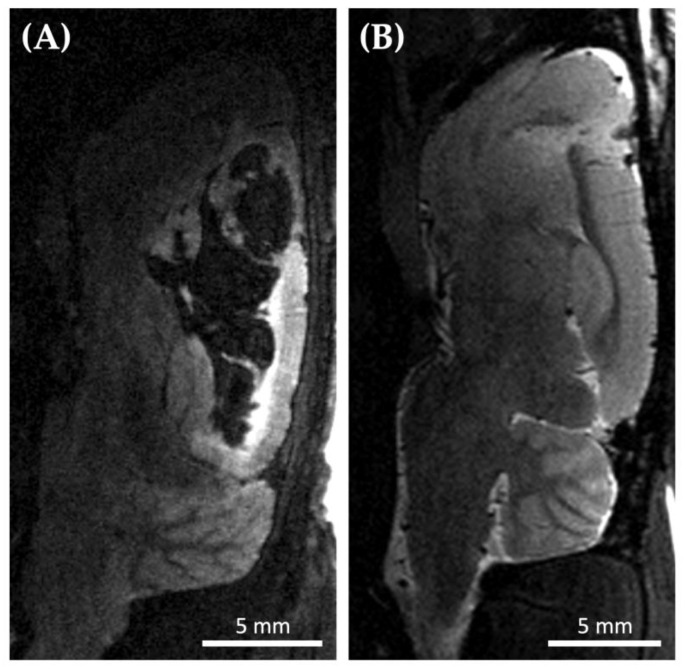
Exemplary brain MRI (Magnetic Resonance Imaging) of two different Wistar rats 21 days after tumor cell inoculation. (**A**) Sagittal T2 weighted MRI of a Wistar rat 21 days after inoculation of C6 tumor cells revealing massive intracerebral hemorrhage (black). Cells were injected right after transfer to the implantation and imaging site (*n* = 4; 3 with significant tumor hemorrhage, 1 with marginal tumor hemorrhage as expected during tumor progression); (**B**) Sagittal T2 weighted MRI of a Wistar rat 21 days after inoculation of C6 cells that were injected 3 days after transfer (*n* = 26). None (*n* = 14) or mild (*n* = 11) hemorrhage occurred in twenty-five of the Wistar rats, while only one rat revealed a moderate tumor hemorrhage.

**Figure 3 biomedicines-11-01518-f003:**
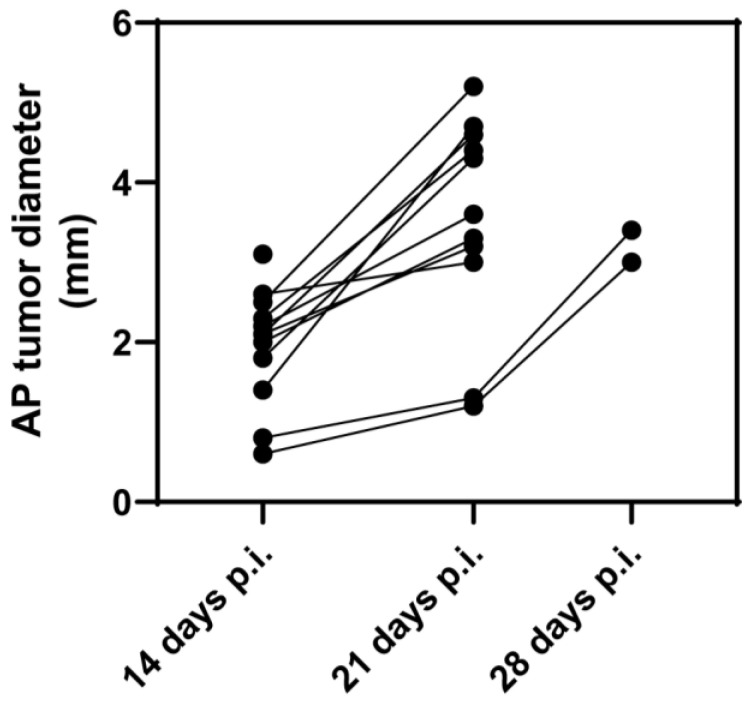
Tumor growth behavior of Wistar rats monitored via MRI (Magnetic Resonance Imaging). Anterior–posterior (AP) tumor diameter measured via contrast-enhancing MRI of Wistar rats (group 1B; animals that underwent partial resection and local drug delivery system implantation and were observed concerning their survival and tumor progression around the implant) 14, 21, and 28 days post-inoculation (p.i.) of C6 tumor cells (*n* = 12). Animals underwent surgery after reaching a tumor diameter of >3 mm: *n*_14d_ = 1; *n*_21d_ = 9; *n*_28d_ = 2.

**Figure 4 biomedicines-11-01518-f004:**
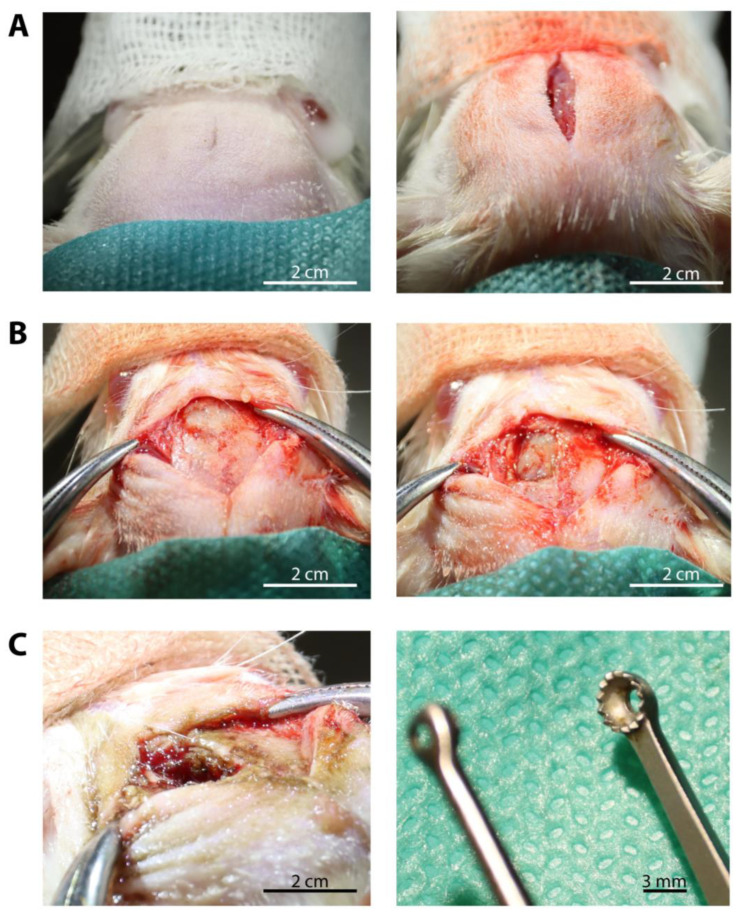
Exemplary pictures taken during the partial tumor resection of a Wistar rat. (**A**) Placement and preparation of the animal and skin incision; (**B**) Exposure of the skull and craniectomy; (**C**) Partial resection of the tumor using alligator forceps.

**Figure 5 biomedicines-11-01518-f005:**
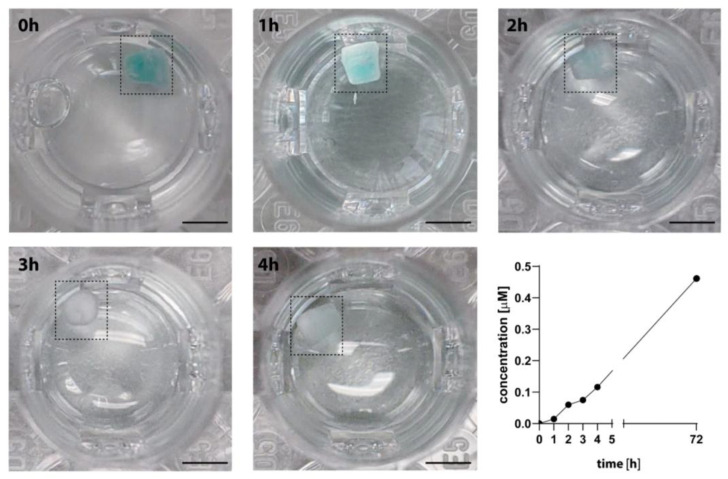
Release of an exemplary local drug delivery system (LDDS) without reservoir in vitro. The LDDS was loaded with 10 μL of a 3.2 μM solution of the IRD 800 dye. Photographs of the loaded LDDS at 0, 1, 2, 3, and 4 h of the release measurement depict the well-visible release of the blue-colored dye out of the implant. The release profile revealed a continuous release of the dye. Bar = 3 mm.

**Figure 6 biomedicines-11-01518-f006:**
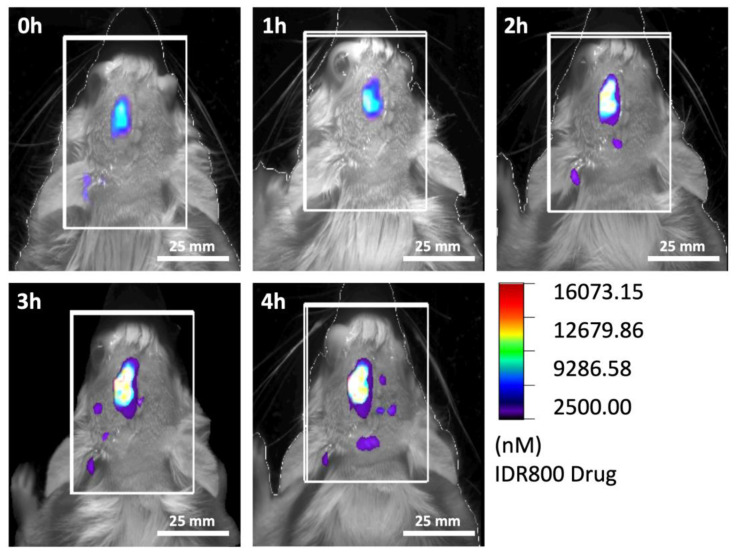
Fluorescence Molecular Tomography (FMT) of a Wistar rat 0, 1, 2, 3, and 4 h after implantation of an exemplary local drug delivery system (LDDS) in a partial glioblastoma resection cavity. The LDDS was loaded with 10 μL of a 3.2 μM solution of the dye IRD 800. FMT showed a steady increase in fluorescence in the resection cavity—likely a sign of a release in vivo.

**Figure 7 biomedicines-11-01518-f007:**
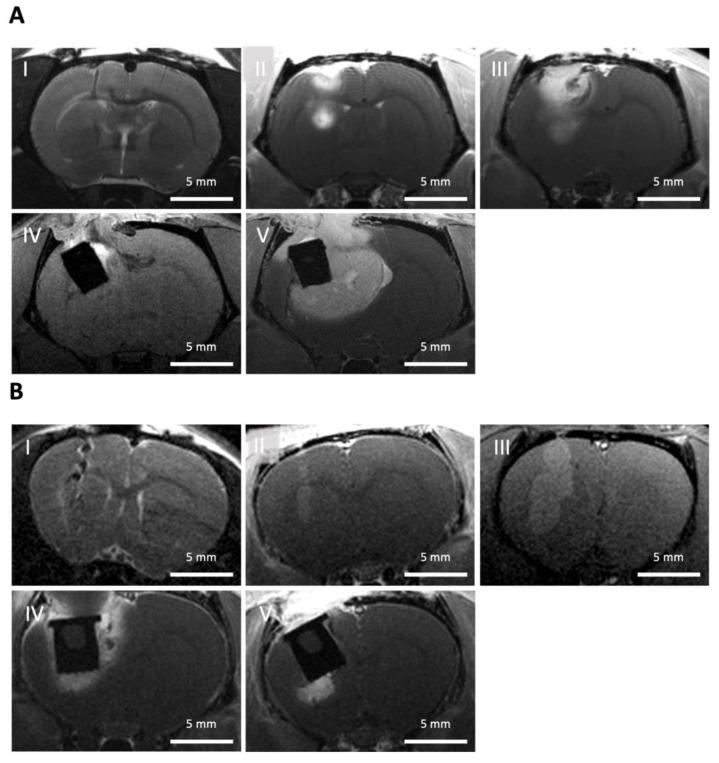
Monitoring of the animals with high-resolution Magnetic Resonance Imaging (MRI). (**A**) Wistar rat: (**I**) Coronal T2-weighted MRI showing the injection path (black) from the inoculation of C6 tumor cells. T1-weighted, post-contrast MRI was used to monitor tumor growth (bright) after (**II**) 14 days and (**III**) 21 days of inoculation. (**IV**) Postoperative MRI revealing the implant (black) loaded with Ringer’s solution and surrounding residual tumor tissue (bright; T1-weighted image). (**V**) Monitoring of tumor progression surrounding the implant (T1-weighted, post-contrast MRI); (**B**) Athymic rat: (**I**) Inoculation of patient-derived glioma stem-like cells (coronal T2-weighted MRI). Tumor growth (**II**) 4 and (**III**) 9 weeks after inoculation, as well as (**IV**) residual tumor tissue after partial resection and introduction of the implant loaded with Ringer’s solution and (**V**) tumor progression around the implant 2 weeks after partial tumor resection (coronal T1-weighted, post-contrast MRI).

**Figure 8 biomedicines-11-01518-f008:**
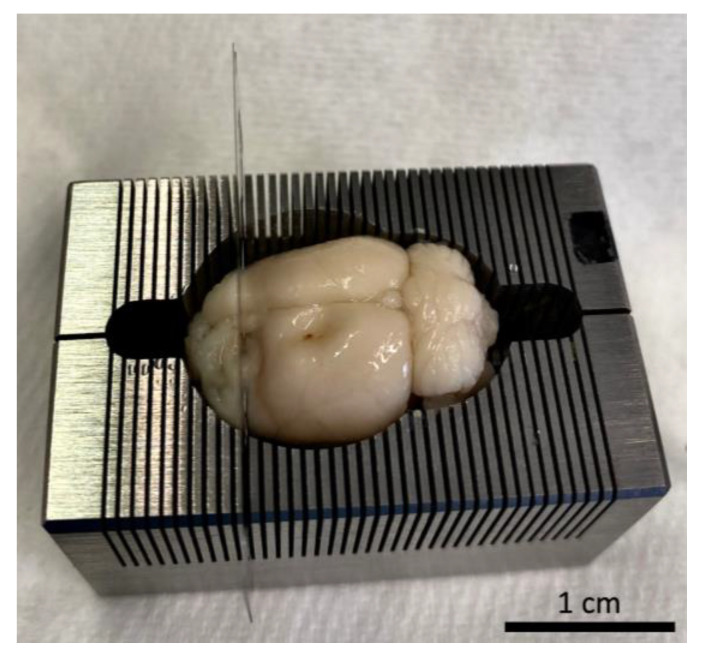
Exemplary picture of the sectioning of a Wistar rat whole-brain using a brain matrix (stainless-steel, coronal slices, 0.5 mm, for rats 300 to 600 g; RBMS-305C, Kent Scientific, Torrington, CT, USA).

**Table 1 biomedicines-11-01518-t001:** Cell counts of viable C6, U87MG, and patient-derived stem-like glioma cells (GSCs) dispersed in Phosphate-buffered saline (PBS) or Ringer’s solution released after initial drawing up of 1 × 10^5^ cells into the Hamilton syringe (26sG). *n* = 4 replicates. * *p* < 0.05.

Cell Line	Cell Count (PBS)	Cell Count (Ringer’s Solution)	Difference in Cell Count PBS/Ringer’s Solution [*n*-Fold]
C6	59,300 ± 1900	55,850 ± 3650	1.07 ± 0.08
U87MG	38,800 ± 2600	41,150 ± 1350	0.94 ± 0.07
Patient-derived GSCs	93,800 ± 1600	96,750 ± 1750	0.97 ± 0.02 *

**Table 2 biomedicines-11-01518-t002:** Cell count of released C6, U87MG, and patient-derived glioma stem-like cells (GSCs) from a 26sG Hamilton needle and a 26G Hamilton needle after initial drawing up of 1 × 10^5^ cells. *n* = 4 replicates. ** *p* < 0.01, and *** *p* < 0.001.

Cell Line	Cell Count 26sG Hamilton Needle (Outer Diameter 0.474 mm, Inner Diameter 0.127 mm)	Cell Count 26G Hamilton Needle (Outer Diameter 0.464 mm, Inner Diameter 0.26 mm)	Difference in Cell Count 26sG/26G Hamilton Needle [*n*-Fold]
C6	61,117 ± 6447	92,175 ± 4007	1.52 ± 0.17 ***
U87MG	39,975 ± 2750	83,367 ± 3691	2.09 ± 0.17 ***
Patient-derived GSCs	45,467 ± 13,943	95,417 ± 6947	2.32 ± 0.73 **

## Data Availability

Not applicable.

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
