# Peer review of "Establishment of a Rodent Glioblastoma Partial Resection Model for Chemotherapy by Local Drug Carriers—Sharing Experience"

_biomedicines, 2023, doi:10.3390/biomedicines11061518_

Round 1

Reviewer 1 Report

The paper entitled "Establishment of a rodent glioblastoma partial resection model for Chemotherapy by local drug carriers - sharing experience" by Carolin Kubelt provides valuable insights into the development of local drug delivery systems (LDDS) for glioblastoma (GBM). The paper presents the results of establishing a GBM partial resection model in rats, which is suitable for evaluating the efficiency of LDDS. The paper highlights the importance of animal models for evaluating the therapy before any human application and emphasizes the need for sharing experiences to benefit from them.

The paper describes the use of MRI to monitor tumor growth and quantify the tumor tissue remaining after resection, which is an important technique for determining the success of LDDS. The paper notes that some LDDS materials may cause MRI artifacts and should be checked first. The paper also suggests that metabolic MRI, such as using hyperpolarized MRI, could be a promising technique for a more detailed picture and early response detection.

The paper highlights the value of fluorescence micro tomography (FMT) in monitoring the release properties of the LDDS used. However, the paper also notes that this approach is limited to small animals, has limited spatial resolution, and the release of the drug may not be the same as the release of the dye. Nonetheless, the information gained from FMT was very valuable, providing a non-invasive measurement for the release properties.

The paper acknowledges that the established rodent GBM partial resection model is complex and cost intensive. However, the paper emphasizes that the demand to resemble human conditions allows for greater transferability into a clinical setting. The paper concludes that the established rodent GBM partial resection model offers a valid basis to test local chemotherapy released by intracavitary-located drug carriers.

I have some minor comments to improve the manuscript, the paper could benefit from including more details in the introduction about the challenges of GBM treatment and the potential of LDDS. Additionally, the paper could benefit from including more data and figures to support the findings presented.

Reviewer 2 Report

 Local drug delivery systems (LDDS) represent a promising therapy strategy concerning the most common and malignant primary brain tumor glioblastoma (GBM). To date, only few systems are clinically applied with limited success, and numerous new LDDS approaches have been currently developed. GBM animal models play a key role as such models and are needed to evaluate the therapy prior to any human application. Such models, however, are complex to establish. In present manuscript, the results of establishing a partial resection glioma model in rats suitable for evaluating LDDS was reported. C6-bearing Wistar rats and U87MG-spheroids- and patient-derived glioma stem-like cells-bearing athymic rats underwent tumor resection followed by the implantation of an exemplary LDDS. Inoculation, tumor growth, residual tumor tissue, and GBM recurrence were reliably imaged using high-resolution Magnetic Resonance Imaging. The release from an exemplary LDDS was verified in vitro and in vivo using fluorescent micro tomography. The presented GBM partial resection model appears to be well suited to determine the efficiency of LDDS. The manuscript is interesting and I have only one minor remark written further.

At the end of introduction part there is need to clearly point out what is the novelty of present research. What was done for the first time?

-

Reviewer 3 Report

This is a very interesting article that touches on an actual modern problem. However, the article has a number of wishes. 1) The introduction should more clearly indicate the purpose of the research in accordance with the results obtained. 2) 2.3. Cultivation of Cells. Rephrase. 3) 3.1. Selection of a suitable animal model and cell lines. I would recommend moving to material and methods or supplement 4) The discussion should be structured into subsections and shortened. 5) A clear conclusion is needed based on the research results at the end of the article

English language needs additional proofreading and small typographical corrections
